# Effects of Variable-Resistance Training Versus Constant-Resistance Training on Maximum Strength: A Systematic Review and Meta-Analysis

**DOI:** 10.3390/ijerph19148559

**Published:** 2022-07-13

**Authors:** Yiguan Lin, Yangyang Xu, Feng Hong, Junbo Li, Weibing Ye, Mallikarjuna Korivi

**Affiliations:** 1Department of Public Instruction, Tourism College of Zhejiang, Hangzhou 311231, China; linyiguan@tourzj.edu.cn; 2Student Affairs Office, Medical College, Shandong Yingcai University, Jinan 250104, China; xuyangyang@sdycu.edu.cn; 3Department of Sports Operation and Management, Jinhua Polytechnic, Jinhua 321000, China; 20201015@jhc.edu.cn; 4Physical Education Department, Zhejiang University of Science and Technology, Hangzhou 310023, China; 100111@zust.edu.cn; 5Institute of Human Movement and Sports Engineering, College of Physical Education and Health Sciences, Zhejiang Normal University, Jinhua 321004, China

**Keywords:** dose–response, training intensity, elastic bands, chain, training load

## Abstract

Greater muscular strength is generally associated with superior sports performance, for example, in jumping, sprinting, and throwing. This meta-analysis aims to compare the effects of variable-resistance training (VRT) and constant-resistance training (CRT) on the maximum strength of trained and untrained subjects. PubMed, Web of Science, and Google Scholar were comprehensively searched to identify relevant studies published up to January 2022. Fourteen studies that met the inclusion criteria were used for the systematic review and meta-analysis. Data regarding training status, training modality, and type of outcome measure were extracted for the analyses. The Cochrane Collaboration tool was used to assess the risk of bias. The pooled outcome showed improved maximum strength with VRT, which was significantly higher than that with CRT (ES = 0.80; 95% CI: 0.42–1.19) for all the subjects. In addition, trained subjects experienced greater maximum-strength improvements with VRT than with CRT (ES = 0.57; 95% CI: 0.22–0.93). Based on subgroup analyses, maximum-strength improvement with a VRT load of ≥80% of 1 repetition maximum (1RM) was significantly higher than that with CRT (ES = 0.76; 95% CI: 0.37–1.16) in trained subjects, while no significant differences were found between VRT and CRT for maximum-strength improvement when the load was <80% (ES = 0.00; 95% CI: −0.55–0.55). The untrained subjects also achieved greater maximum strength with VRT than with CRT (ES = 1.34; 95% CI: 0.28–2.40). Interestingly, the improved maximum strength of untrained subjects with a VRT load of <80% of 1RM was significantly higher than that with CRT (ES = 2.38; 95% CI: 1.39–3.36); however, no significant differences were noted between VRT and CRT when the load was ≥80% of 1RM (ES = −0.04; 95% CI: −0.89–0.81). Our findings show that subjects with resistance training experience could use a load of ≥80% of 1RM and subjects without resistance training experience could use a load of <80% of 1RM to obtain greater VRT benefits.

## 1. Introduction

Maximum strength is the maximum force a muscle can generate in a single isometric voluntary contraction [1]. The performance of athletes, especially in powerlifting and weightlifting, is directly associated with their maximum strength. Athletes in sports such as track-and-field, wrestling, and basketball also require maximum strength for better performance [2,3]. Constant-resistance training (CRT) is a type of training that uses constant weight loads to improve the maximum strength of an individual [4]. However, CRT does not produce effective muscle stimulation over the entire range of motion because of the “sticking point” [5,6,7]. Variable-resistance training (VRT), also called accommodating-resistance training [8], uses an elastic band or chain and is an alternative training method to CRT. VRT facilitates different weight loads and helps to overcome the sticking point during resistance training. VRT can reduce skeletal muscle resistance in the weakest area of motion, provide greater resistance in areas with more strength, and get closer to human strength curves to make the muscles function over a broader range [9]. As a result, VRT has the potential to increase motor unit recruitment and firing rates and improve training benefits [10,11,12]. Many studies have shown that VRT is effective in improving maximum strength [13,14,15]. However, there is inconsistent evidence to support this hypothesis [16]. In addition, VRT has been shown to produce greater stimulation of muscles during the eccentric phase, thereby increasing the rate of force development and obtaining a greater muscle stretch–shortening cycle [17,18,19]. The training benefits of VRT are associated with neuromuscular adaptations. VRT can activate muscle fibers to participate in contractile movement to a greater extent [20,21]. VRT produces appropriate instability in the exercise and keeps muscles in a state of tension during the eccentric phase, which can help athletes recover from injuries. Therefore, VRT is beneficial in post-operative rehabilitation [4].

With the increase in research on VRT, contradictory research data have emerged. The results of several of studies have not found that VRT is better than CRT for the development of maximum strength [16,22,23]. In a study by Cronin et al. [24], participants performed supine jump squat training with a load of 8–15 repetition maximum (RM) with or without elastic bungees. The results revealed that maximum strength of participants with elastic bungees was not better than that of participants without bungees (non-bungee squat) [24]. In a similar study, participants used a combination of chains and without chains for jump squat training with a 30% 1RM load. The results also showed that VRT did not effectively improve maximum strength [24]. Ebben et al. [25] also showed that there were no significant differences in neuromuscular activation between VRT and CRT through electromyography (EMG) of the hamstrings and quadriceps.

Two recent meta-analyses attempted to address the influential role of VRT over CRT on gaining of muscular strength in different populations. These two studies reported no significant differences in the development of maximum strength between VRT and CRT [26,27]. Furthermore, these studies [26,27] were limited with a smaller number of included articles, inadequate details of subjects/training loads, lack of subgroup analysis, and results seem to be inconsistent with the widely held view. Thus, whether VRT contributes to maximum-strength improvement and quantifying the dosage of appropriate exercise for optimal strength are problems that need clarification [28]. Based on current reviews, the VRT development of maximum strength is still controversial, so further analysis is needed to unequivocally determine the effects of VRT. The purpose of this study was to verify the impact of VRT on maximum strength and to analyze the factors that limit the beneficial effects of VRT on improving maximum strength. The hypothesis was that the effects of VRT and CRT on maximum strength are the same.

## 2. Materials and Methods

### 2.1. Search Strategy

We followed the Preferred Reporting Items for Systematic Reviews and Meta-Analyses (PRISMA, 2020) guidelines [29] (Appendix A). We searched the databases for relevant articles up to 31 January 2022 without restricting the starting date. The literature retrieval was carried out independently by researchers Y.L. and W.Y. The articles included in our study were obtained by searching for randomized and non-randomized controlled trials published in English. Articles related to variable-resistance training were searched in the PubMed, Web of Science, and Google Scholar databases using combinations of the following keywords: variable-resistance training; accommodating resistance; chain training; elastic training; rubber band; maximum strength; compensatory acceleration training; squat, bench press; barbell deadlift. The database search was limited to peer-reviewed English journal articles. After retrieving the publications, the reference lists were searched twice for a more comprehensive inclusion of other articles of potential interest.

### 2.2. Inclusion and Exclusion Criteria

Before inclusion, the searched articles and abstracts were screened; then, the full text of each article was obtained. A strict review was then conducted in accordance with the inclusion criteria. The analysis did not limit the subjects’ age, sex, training basis, sports specialty, or body composition. Inclusion criteria were as follows: (1) at least one group in the experiment was trained in variable-resistance mode; (2) the outcome measure was maximum strength; (3) the study was published in a peer-reviewed English journal.

Studies were excluded if (1) the maximum-strength index was not reported in the experiment; (2) the studies included only an abstract without full text; (3) the studies did not provide sufficient outcome data; (4) studies were duplicate publications; (5) no comparisons of the effects before and after training modes were conducted; and (6) there was a lack of a CRT group.

The articles were independently screened by two investigators (Y.L. and W.Y.) according to the inclusion and exclusion criteria. By reading the abstracts and text, articles that did not conform to our requirements were excluded. We then continued reading the full text of articles that met the inclusion criteria. If investigators’ opinions were not unanimous, another review author (M.K.) was invited to negotiate and reach a consensus.

### 2.3. Quality Evaluation

The Cochrane Collaboration tool was used to determine the risk of bias for the included trial as described in the handbook [30]. The included full-text articles were assessed by two of the three review authors (Y.L., Y.X., and W.Y.), and the risk-of-bias tool was independently applied to each study. The differences were resolved by discussing with other review authors (J.L. and M.K.). Sources of biases, such as selection bias (random sequence generation and allocation concealment), performance bias (blinding of participants and personnel), detection bias (blinding of outcome assessment), attrition bias (incomplete outcome data), and reporting bias (selective reporting) were detected for all the included studies. The outcome of the risk of bias is fully described in Section 3, Results.

### 2.4. Data Extraction

The basic information from the articles that met our criteria, including authors; sex, age, and number of participants; training basis; training methods; training arrangements (training cycle, number of weekly training sessions, number of groups, number of repeats); and load, is presented in Table 1. This task was undertaken by one author (Y.L.). The second and third authors (Y.X. and F.H.) checked the extracted data for accuracy and completeness. A quality assessment was conducted by another review author (W.Y.). Disagreements were resolved by consensus or by another author (M.K.).

### 2.5. Outcome Measures

All studies used maximum strength as the evaluation indicator. The maximum-strength index was measured using a barbell for the 1RM in kilograms (kg). The maximum strength of the subjects was tested before and after training, and the change in maximum strength before and after training was measured.

### 2.6. Data Analysis

We employed the Cochrane Collaboration Review Manager (RevMan, Copenhagen, Denmark) version 5.3 for the statistical analyses. The I^2^ test was used to test the heterogeneity of each trial, and 25%, 50%, and 75% of the values represented low, medium, and high statistical heterogeneity. If there were no significant differences in the heterogeneity test, a fixed effects model was employed for the meta-analysis; a random effects model was used when there was high heterogeneity in the heterogeneity test. For continuous outcome variables with the different test units and methods, the standardized mean difference (SMD) and 95% confidence intervals (CIs) were selected as the effect sizes for the combined analysis. Meta-regression analysis was performed for VRT duration and load to identify their influential role on gaining maximum strength. Then, subgroup analysis was performed to determine the optimum load of VRT that could effectively improve maximum strength.

Meta-analysis data were extracted from the change values of the VRT and control groups before and after the intervention, namely, the mean ± SD of the change values before and after training. When relevant data were unavailable, the filling method was adopted based on the research study by Bellar et al. [31], and a correlation coefficient of 0.986 was obtained. Based on the correlation coefficient, the SD changes before and after training in the remaining included articles were obtained. The calculation formula was as follows [30]:$${\mathrm{SD}}_{\mathrm{change}}=\sqrt{{\left[SD_{pre}\right]}^{2}+{\left[SD_{post}\right]}^{2}-2\times corr\times SD_{pre}\times SD_{post}}$$
where SD_change_ is the standard deviation of change values before and after training; *SD_pre_* is the standard deviation before training; *SD_post_* is the standard deviation after training; corr is the correlation coefficient.
ijerph-19-08559-t001_Table 1Table 1Details of the studies included in the meta-analysis.Study*n*SexAge (Years)ExperienceTraining MethodsTraining ArrangementIntensity (%)VRTCRTVRTCRT
PMRPVRPCRSawyer et al. 2021 [32]2020Male18–25TrainedSquat + elasticSquat3 × 3w [5 × (1–7)]50–932080Arazi et al. 2020 [33]1212Female24 ± 4UntrainedSquat + chainSquat3 × 8w [(3–5) × (6–12)]65–8515851212Female24 ± 4UntrainedBench press + chainBench press3 × 8w [(3–5) × (6–12)]65–851585Kashiani et al. 2020 [34]1716Male22 ± 2UntrainedOverhead press + chainOverhead press3 × 12w [3 × (8–12)]70–8035651716Male22 ± 2UntrainedOverhead press + elasticOverhead press3 × 12w [3 × (8–12)]70–803565Katushabe et al. 2020 [35]98Male21 ± 2TrainedSquat + elasticSquat— × 6w [3 × (5–10)]-208098Male21 ± 2TrainedDeadlift + elasticDeadlift— × 6w [3 × (5–10)]-2080Archer et al. 2016 [24]1110Male24 ± 2TrainedSquat jump + chainSquat jump3 × 1w [5 × 3]302080Anderson et al. 2015 [22]1616Female24 ± 6TrainedSquat + elasticSquat2 × 10w [(3–4) × (6–10)]75–8527–5842–73Ataee et al. 2014 [8]88Male21 ± 2TrainedSquat + chainSquat3 × 4w [1 × 5]85208088Male21 ± 2TrainedBench press + chainBench press3 × 4w [1 × 5]852080Bellar et al. 2011 [31]1111Male24 ± 3UntrainedBench press + elasticBench press2 × 13w [5 × 5]851585Shoepeet al. 2011 [16]1011Mixed20 ± 1UntrainedBench press + elasticBench press3 × 24w [(3–6) × (6–10)]67–9520–3565–801011Mixed20 ± 1UntrainedSquat + elasticSquat3 × 24w [(3–6) × (6–10)]67–9520–3565–80Burnham et al. 2010 [36]109Female20 ± 2TrainedBench press + chainBench press2 × 8w [3 × (4–6)]80–90595Ghigiarelli et al. 2009 [37]1212Male20 ± 1TrainedBench press + elasticBench press4–5 × 7w [(5–6) × (4–6)]85--1212Male20 ± 1TrainedBench press + chainBench press4–5 × 7w [(5–6) × (4–6)]85--McCurdy et al. 2009 [23]1312Male21 ± 1TrainedBench press + chainBench press2 × 9w [(5–7) × (5–10)]60–9510–2080–90Rheaet al. 2009 [38]1616Male21 ± 2TrainedSquat + elasticFast squat2–3 × 13w [4 × 10]75–85--1616Male21 ± 2TrainedSquat + elasticSlow squat2–3 × 13w [4 × 10]75–85--Anderson et al. 2008 [39]2321Mixed20 ± 1TrainedBench press + elasticBench press3 × 7w [(3–6) × (2–10)]72–9820802321Mixed20 ± 1TrainedSquat + elasticSquat3 × 7w [(3–6) × (2–10)]72–982080Note: The content of the study design comprises training times per week × training weeks [(sets) × (repetitions)], excluding warm-up and relaxation. VRT = variable-resistance training; CRT = constant-resistance training; w = week; PMR = percentage of maximum repetitions; PVR = percentage of variable resistance; PCR = percentage of constant resistance; *n* = number of participants.

## 3. Results

### 3.1. Search and Exclusion Results

Following a systematic search, we retrieved a total of 2436 articles. After removing the duplicate records (1132), 331 records were marked as ineligible by automation tools, and 467 records were removed for other reasons. From the remaining (506) records, 471 were excluded according to our study criteria, leaving 35 articles. Finally, there were 35 articles relevant to our study. The remaining 35 articles were further evaluated, and 21 were screened out for the following reasons: 3 studies did not report average or standard deviations [17,40,41]; 1 study did not report maximum-strength indicators [42]; 15 studies did not compare the effects before and after the training intervention [15,20,21,25,43,44,45,46,47,48,49,50,51,52,53]; 2 studies had no CRT group [54,55]. Finally, a total of 14 studies were included in the meta-analysis. The specific screening steps are shown in Figure 1.

### 3.2. Description of Included Studies

In our systematic review and meta-analysis, we included 14 studies comprising 22 reports. These studies were intercontinental, and published between 2008 and 2021. The reports involved 414 participants (trained and untrained) with a mean age between 18 and 30 years. The specific details are shown in Table 1. Of the included studies, three studies only recruited female participants, nine only involved male participants, and two involved male and female participants. In terms of training, four studies were conducted on untrained subjects, and 10 studies were conducted on trained subjects. The main training methods were squatting and bench pressing. The VRT forms included chain and elastic resistance combined with barbells. In terms of the training period, 10 studies were ≤ 10 weeks, and four studies were >10 weeks. The percentage of maximum repetitions was from 30 to 95%; the proportion of the variable load component accounted for 10–35% of the total load.

### 3.3. Meta-Analysis Results of VRT and CRT Modes on Maximum Strength

A total of 22 reports that comprised both trained and untrained participants were included for the meta-analysis [8,16,22,23,24,31,32,33,34,35,36,37,38,39]. As shown in Figure 2, VRT and CRT significantly differed in the improvement of the maximum strength of the subjects (ES = 0.80; 95% CI: 0.42–1.19). However, high statistical heterogeneity (I^2^ = 78%) was detected in our analysis.

### 3.4. Influence of VRT and CRT on Maximum Strength of Trained Subjects

The meta-analysis conducted on the studies of only trained subjects showed that VRT favored a significantly higher improvement of maximum strength than CRT (ES = 0.57; 95% CI: 0.22–0.93; Figure 3) [8,22,23,24,32,36,37,38,39]. Based on the VRT workload, we then subgrouped the studies into <80% and ≥80% 1RM. As reported in Figure 3, the effect of VRT with a load of ≥80% 1RM on the maximum-strength development was significantly higher than that of CRT (ES = 0.76; 95% CI: 0.37–1.16). However, no significant differences were observed between VRT and CRT in the improvement of maximum-strength when the load of VRT was <80% 1RM (ES = 0.00; 95% CI: −0.55–0.55; Figure 3).

### 3.5. Influence of VRT and CRT on Maximum Strength of Untrained Subjects

As shown in Figure 4 [16,31,33,34], maximum-strength gains were significantly higher with VRT than CRT in the untrained subjects (ES = 1.34; 95% CI: 0.28–2.40). Interestingly, the subgroup analysis showed that the effect of VRT with a load of <80% 1RM on maximum-strength gain was significantly greater than that of CRT (ES = 2.38; 95% CI: 1.39–3.36). Nevertheless, we found no significant differences between VRT and CRT in the development of maximum strength when the load of VRT was ≥80% 1RM (ES = −0.04; 95% CI: −0.89–0.81) in the untrained subjects (Figure 4).

### 3.6. Risk of Bias in the Results

We used the Cochrane collaborative method to assess the risk of bias (Figure 5) [8,16,22,23,24,31,32,33,34,35,36,37,38,39]. For the selection bias, 12 trials reported random sequence generation and two non-randomized groupings; no reports of concealment and blinding were documented, and all the literature was rated as having a high risk. The outcome variable evaluation was not mentioned in 14 studies, and all articles were assessed as being unclear. In our evaluation, one study identified a reporting bias. No experiments indicated follow-up bias or other biases.

## 4. Discussion

To the best of our knowledge, this is the first systematic review and meta-analysis to investigate the effect of VRT and VRT load on maximum-strength gain in comparison with CRT. Our results show that VRT was better than CRT in improving the maximum strength of trained and untrained subjects. Furthermore, the VRT-improved maximum strength depended on the workload. The subgroup analysis showed that the VRT beneficial effect was better when the untrained subjects used a <80% 1RM load and when the trained subjects used a load of ≥80% of 1RM.

Many studies have indicated that a ≥80% 1RM load is the most conducive to developing muscle strength and have used this as the boundary value of the load [2,56]. During strength training, as the resistance increases, the speed of the movement gradually decreases, resulting in a “sticking region” at the weakest position of the joint. When athletes use CRT in heavy-load training, they often fail to lift weights in the concentric phase because of the sticking region of their movements, thus reducing the degree of stimulation produced by training on the target muscles. When VRT is used, the resistance of weak muscle points is reduced, which, in turn, reduces the probability of weight lifting failure. At the same time, the resistance gradually increases in the latest stage of the action and exceeds the maximum weight that could be lifted when CRT is used, thus producing greater stimulation of the target muscles. Therefore, it is likely that the concentric stage of VRT is the most favorable component to facilitate the development of maximum strength, especially in the latest stage of the concentric action [57]. Israetel et al. [15] showed, using EMG, that in the squat movement, the activation of vastus lateralis was the highest in the early stage of the concentric phase and late stage of the eccentric phase under VR conditions. During squat and bench presses, VRT is able to provide progressive resistance to match the human strength curves [4]. The early stage of the concentric phase and the late stage of the eccentric phase are the stages in which the greatest resistance occurs, and stimulation with a heavy load is necessary to increase strength.

This meta-analysis shows that trained subjects obtained a better effect from VRT with a training load ≥80% of 1RM. When training with a smaller load, the load does not reach the limit of muscle strength, and there is no sticking region [7]. At the same time, the muscles do not bear the overload at the latest stage of the movement. VR is not enough to stimulate the growth of strength to a great extent, and the benefits brought by VR are reduced. For trained subjects, less than 80% 1RM loads did not properly stimulate the muscles, so the training effect of VRT was not significant compared with CRT. However, movement speed may be a factor in maximum strength. Rhea et al. [38] found that the increase in maximum strength was more significant with slow training. This may be related to slower training during the concentric phase contributing to the increased cross-sectional area of type I and type II-a skeletal fibers [46]. In general, when the load is small, VRT may trigger higher movement speeds, which affects the increase in maximum strength to some extent [43,44,46]. Cronin et al. [41] and Archer et al. [24] both used lower loads for power training, and the results showed that the VRT group had a lower maximum-strength increase than the CRT group. However, Stevenson et al. [47] argued that VRT can increase the speed of the eccentric phase but can harm the speed of the concentric phase. Recent analyses of the mechanism by which VRT increases maximum force revealed that the speed increase in VRT mainly occurred during the eccentric phase and that eccentric acceleration may contribute to the maximum increase in strength [57,58].

Another meta-analysis concluded that a <80% 1RM VRT load had a more significant effect on untrained subjects. Strength improvement mainly depends on muscle and nerve adaptation. Muscle adaptation includes improved energy reserves, increased muscle fiber size, and capillary density. Nerve adaptation includes the activation of motor units, intermuscular coordination, and changes in the discharge frequency of motor neurons [59]. Hakkinen et al. [60] found that the first 8 weeks of strength training mainly improved nerve adaptability, while the second 8 weeks increased the muscle fiber size. Several studies have suggested that the maximum strength increased by VRT is mainly related to improvements in neuromuscular adaptation [20,39]. For trainers who have had long-term strength training, their power may increase to a higher level. However, it becomes complicated to increase other muscles’ sizes and strength. Adding a chain or elastic band to the free weights or changing the state of the body movement can provide a new stimulus for the muscles and improve the coordination between the muscles in the fight against unfixed resistances, thus improving the development of strength. Mina et al. [21] compared the effects of VRT and CRT on post-activation potentiation (PAP). Their results showed that warming up with VRT was more beneficial in improving subsequent 1RM performance. A recent study by Smith et al. [9] reported that VRT showed shorter electrochemical (reflex-EMDE-M) and mechanical (reflex-EMDM-F) activities after four weeks of training. These studies also support the opinion that VRT training can improve neuromuscular adaptation. For individuals with no training experience, strength enhancement is mainly based on neural adaptation; the mobilization ability of muscles is weak, and the excessive load may not produce optimal stimulation of muscles. Therefore, it is more appropriate to use a load of <80% of 1RM for VRT.

In VRT, the ratio of VR to CR is also an aspect worth exploring. Some research groups have suggested that if the goal is to develop maximum strength, training with VR accounts for 15–35% of the total resistance and produces a better training effect [17,39,61]. If the purpose of training is to develop explosive force, a VR load accounting for 10–20% of the total resistance should be used for training [44,61]. In two studies, 80% of 1RM (5% VR, 75% CR) and 85% of 1RM (5% VR, 80% CR) were used to carry out a comparative study of Olympic clean and snatch exercises. The results revealed that there were no significant differences between the force output of Olympic clean and snatch in the VRT form and that of CRT. The subjects also reported that it was more challenging to carry out Olympic clean and snatch exercises in the VRT form [13,50]. Therefore, the impact of different training actions should also be considered in the implementation of training regimes, which is again a topic that requires further research.

There are a few limitations to the present study. First, due to the lack of necessary data, several relevant studies were excluded, which resulted in a relatively lower number of included studies. Second, there were no allocation concealment and blinding methods in the studies’ quality evaluations, so the risk of bias was high. Third, some of the loads used in the included studies were varied. In our study, an intermediate load value was considered for the total load, but there may have been some deviation. Despite these limitations, overall, our meta-analysis provides new ideas and conclusions that emphasize the beneficial effects of VRT and its load on improving the maximum strength of trained and untrained athletes.

## 5. Conclusions

Our findings suggest that VRT is better than CRT in improving maximum strength. Trainers of different levels should choose the corresponding load of VRT for training according to the actual situation. Our findings recommend that trained subjects could use a load of ≥80% of 1RM and that untrained individuals could use a load of <80% of 1RM to attain greater VRT benefits. Future research should refine the training load, such as distinguishing between different-level trainers using the training benefits of VRT with different loads, and the proportion of variable resistance in the total load. In addition, the specific differences in the training effects of the two forms of VRT, the iron chain, the elastic band, and the impact of the training cycle are still worthy of further research.

## Figures and Tables

**Figure 1 ijerph-19-08559-f001:**
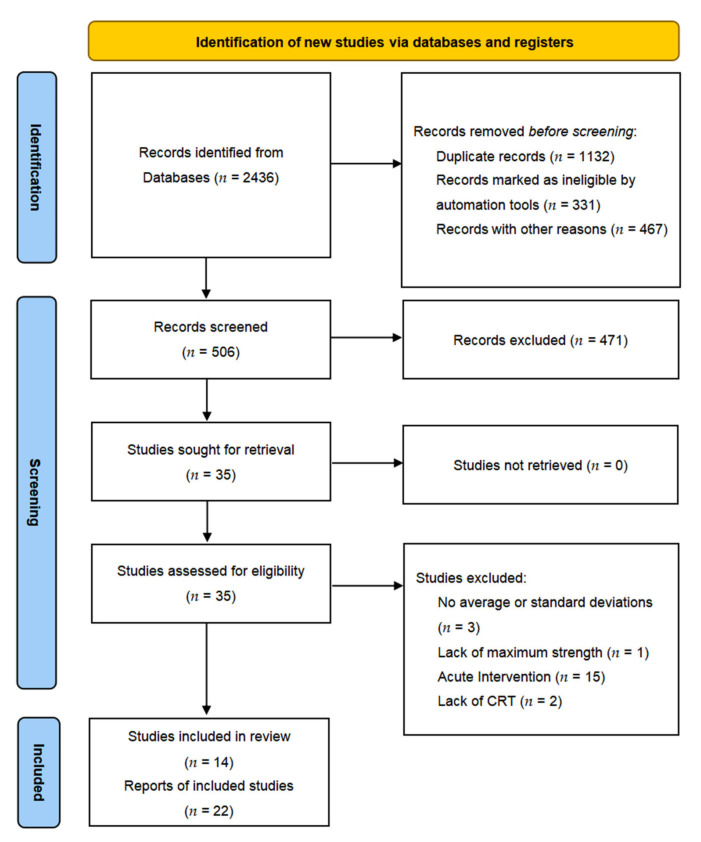
Preferred Reporting Items for the Systematic Review and Meta-Analysis (PRISMA) flow diagram of article selection.

**Figure 2 ijerph-19-08559-f002:**
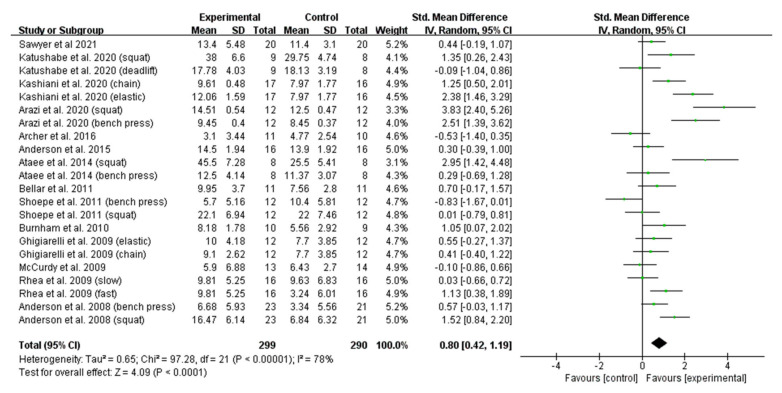
Forest plot of maximum-strength development comparison between VRT and CRT.

**Figure 3 ijerph-19-08559-f003:**
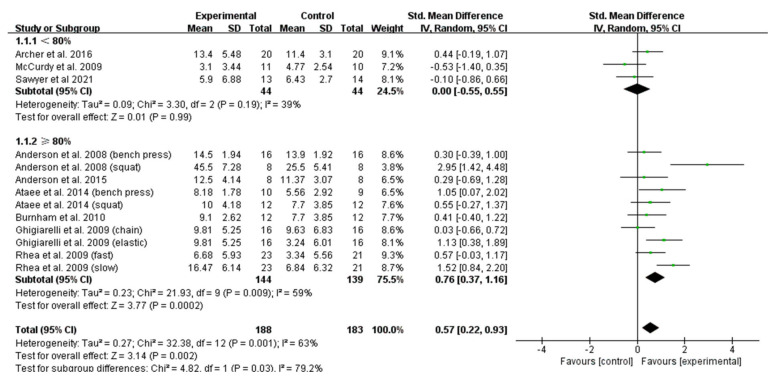
Forest plot of maximum-strength development: comparison between VRT and CRT after sensitivity analysis in trained subjects.

**Figure 4 ijerph-19-08559-f004:**
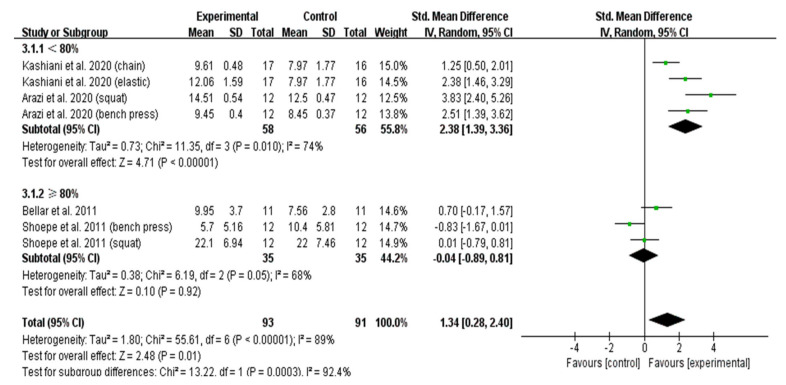
Forest plot of maximum-strength development: comparison between VRT and CRT after sensitivity analysis in untrained subjects.

**Figure 5 ijerph-19-08559-f005:**
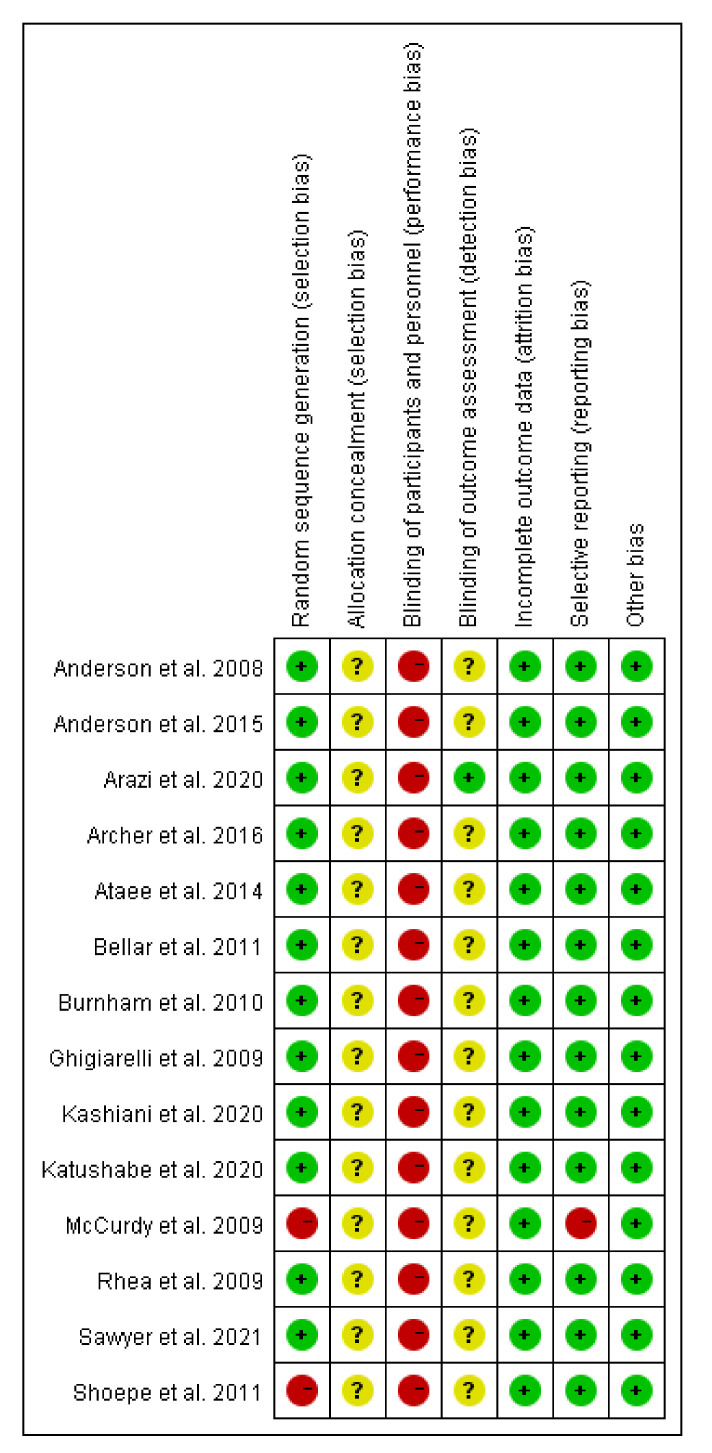
Summary of the risk of bias of studies included in this meta-analysis. Green indicates a low risk of bias, yellow indicates unclear bias, and red indicates a high bias risk.

## Data Availability

Not applicable.

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
