# Peer review of "Effects of Variable-Resistance Training Versus Constant-Resistance Training on Maximum Strength: A Systematic Review and Meta-Analysis"

_ijerph, 2022, doi:10.3390/ijerph19148559_

Round 1

Reviewer 1 Report

I would like to express my gratitude regarding the opportunity to review this manuscript.

It is an interesting study, congratulations, but at this stage still requiring many improvements. Below suggestions/indications with page and line indication:

L 6-15 - Please consider providing the institutional emails.

L 16-17 - Please consider phrase reformulation since many other factors influence and determine sports performance.

L 41 - In this line and throughout all the manuscript a space should be presented before citations “[“.

L 70 - “1RM” should be in full in the first appearance in the text.

L 83 - VRT e CRT not necessary in full, already previously in the beginning of introduction section.

L 88 - End point in red, please correct.

L 98 - Please indicate the time frame of research, the beginning date information is not it the text.

L 118 - “(23)” It is not understandable what represents. Please review.

L 121 - Space before author indication.

L 138 - 1RM should be previously in full and in this line only “1RM” (please see line 70).

L 141 – Please consider quality assessment, for example with PEDro methodological quality.

Please carefully review table 1 content. For example “Trainingarrangement” and years of publication should follow journal guidelines “[ ]”.

Page 7 - Please consider figure format (type and size of letter according to instructions for authors) and size of the boxes, aiming that the figure is placed in the same page of the 3.1 section, to provide better reading and interpretation.

Page 8 - L 34 - Please consider improving figure quality, journal guidelines “[ ]” and values with the same decimal format. The same in Page 9, Figures 3 and 4.

Page 10 figure should be configured assuming what was previously indicated for page 7 and figure should present title and legend.

P14 - L94-120 - The paragraph is too long, please review

Page 11 L 133 - Word track changes is present, please review.

Page 12 - L169-171 - Please consider improving the results section, only 3 lines.

Page 12 L 172 - Supplementary materials. Consideration should be given to providing the Prisma checklist output.

Page 12 - “Acknowledgments:” missing

English should be carefully reviewed throughout the manuscript.

References format should be carefully reviewed and corrected. They are not currently according to the journal instructions.

Author Response

Reviewer 1

Dear Reviewer

We appreciate the critical evaluation and meaningful comments of the Reviewer on our manuscript. All comments helped us to improve the quality of our manuscript. We have taken special care to revise the entire manuscript based on the comments. Furthermore, additional information was provided wherever necessary. All corrections in the revised manuscript were marked in red color. Now our detailed 'point-by-point' responses to each comment have been provided below. We believe the revised manuscript is more suitable for your valuable decision.

Comment 1: It is an interesting study, congratulations, but at this stage still requiring many improvements. Below suggestions/indications with page and line indication:

L 6-15 - Please consider providing the institutional emails.

Response: Authors are thankful to the reviewer for his/her greetings. As suggested, now we have provided the institutional email ids in the Title page L 7-15.

Comment 2: L 16-17 - Please consider phrase reformulation since many other factors influence and determine sports performance.

Response: Thank you for your suggestions, now we have rephrased the sentence in line 17-18.

Comment 3: L 41 - In this line and throughout all the manuscript a space should be presented before citations "[".

Response: We have checked throughout the manuscript for the space before citations "[", and corrected as suggested.

Comment 4: L 70 - "1RM" should be in full in the first appearance in the text.

Response: Thank you for your suggestion, now have fully defined the 1RM in the Abstract, L29.

Comment 5: L 83 - VRT e CRT not necessary in full, already previously in the beginning of introduction section.

Response: We agree with reviewer’s opinion that no needs to explain the abbreviations twice. We have deleted the repeated full forms of VRT and CRT in L88-89.

Comment 6: L 88 - End point in red, please correct.

Response: Thank you for this comment. We have now corrected it.

Comment 7L 98 - Please indicate the time frame of research, the beginning date information is not it the text.

Response: We would like to bring to your kind notice that we haven’t set the start date for the article search.

We searched the articles from various databases without restricting the starting date.  Now this information was included in the revised manuscript, Page 2, L 93-94.

Comment 8 L 118 - "(23)" It is not understandable what represents. Please review.

Response: Thanks for this comment. We mean to say that we have followed the handbook of Cochrane to assess the risk of bias. Now we have corrected the citation and sentences for better understanding in the revised manuscript, Page 3, L 125-126.

Comment9 L 121 - Space before author indication.

Response: We have checked this in the whole manuscript and corrected wherever applicable.

Comment 10L 138 - 1RM should be previously in full and in this line only "1RM" (please see line 70).

Response: Thank you for this comment. As we fully explained the 1RM abbreviation earlier, here we deleted the full form of 1RM as suggested.

Comment 11L 141 – Please consider quality assessment, for example with PEDro methodological quality.

Response: We are highly thankful to the reviewer for this comment. We are apologizing that we couldn’t do the PEDro methodological quality for this time. Instead, we assessed the quality of studies using Cochrane Collaboration tool.   

Comment 12 Please carefully review table 1 content. For example Comment "Trainingarrangement" and years of publication should follow journal guidelines "[ ]".

Response: We appreciate Reviewer for his/her careful evaluation of the manuscript. As suggested, we have carefully reviewed the Table 1, and corrected several items, including ‘training management and reference format.

Comment 13Page 7 - Please consider figure format (type and size of letter Comment figure is placed in the same page of the 3.1 section, to provide better reading and interpretation.

Response:  Thank you for your suggestions. We have changed the size of figure and figure legend for smooth reading and better appearance. During the file conversion (from word to PDF) the figures may be displaced a little bit, but the figures in the final version of the paper will be placed accurately with appropriate figure legends. 

Comment 14: Page 8 - L 34 - Please consider improving figure quality, journal guidelines "[ ]" and values with the same decimal format. The same in Page 9, Figures 3 and 4.

Response: We are grateful to the Reviewer for this suggestion. As suggested, now we have improved the quality of all Figures, including Figure 2, 3 and 4 from Page 7 to 9. Furthermore, the decimal points were corrected and maintained the consistency. The references style also corrected according to the journal guidelines. 

Comment 15Page 10 figure should be configured assuming what was previously indicated for page 7 and figure should present title and legend.

Response: We appreciate Reviewer for his/her critical evaluation of our manuscript. As pointed out, we have revised the Figure 5 in Page 9, and this figure was in consistent with data mentioned in Page 6.

                We are apologizing for missing the Figure legend in our earlier version of the manuscript. Now we have included the Figure legend for Figure 5, in Page 9 of the revised manuscript. .

Comment 16P14 - L94-120 - The paragraph is too long, please review

Response: Authors are thankful to the reviewer for this comment. As suggested, we have shortened the paragraph length and revised it for better understanding. P10, L277 - 294

Comment 17: Page 11 L 133 - Word track changes is present, please review.

Response: Thank you for pointing out this correction. Now we have corrected it in the revised manuscript.

Comment 18: Page 12 - L169-171 - Please consider improving the results section, only 3 lines.

Response: Authors are thankful to the reviewer for this comment. We have expanded our conclusions and fully explained our research perspectives in the revised manuscript, Page 11.  

Comment19: Page 12 L 172 - Supplementary materials. Consideration should be given to providing the Prisma checklist output.

Response: Authors are thankful to the reviewer for this meaningful comment. Now we are providing the PRISMA checklist output as supplementary materials.

Comment 20: Page 12 - "Acknowledgments:" missing

Response: Thank you for this comment. Now we have included the "Acknowledgments" in the revised manuscript, Page 11. 

Comment 21: English should be carefully reviewed throughout the manuscript.

Response: As suggested the revised version of the manuscript has been edited by MDPI English editing service. The certificated is attached in the bottom of this response file.

Comment 22: References format should be carefully reviewed and corrected. They are not currently according to the journal instructions.

Response: Thank you for this useful suggestion. We carefully checked all the References and the style formatted according to the journal guidelines. 

Reviewer 2 Report

VRT, a neuromuscular training, has recently been used in many sports training fields. It is meaningful to present standards for effective VRT training.

The expected outcome is that VRT training will be more effective than CRT training. The researcher is presenting the results based on 80% strength, and I want to add a part about the training period and gradual load increase.

If it is difficult to provide evidence for this, it should be explained as a limitation of the study.

In the discussion, it may be effective for muscle training that CRT use can give greater impetus to muscles. This should be explained.

The study was well written according to the procedure, but the conclusion was too simplified. It is necessary to present complex results based on valid reasons why VRT training is more effective.

Author Response

Reviewer 2

Dear Reviewer

We appreciate your critical evaluation and meaningful comments on our manuscript. Your comments helped us to improve the quality of our manuscript. We have taken special care to revise the whole manuscript, additional information provided wherever necessary. All corrections in the revised manuscript were marked in red color. Now our detailed 'point-by-point' responses to each comment have been provided below. We believe the revised manuscript is more suitable for your valuable decision.

Comment 1: VRT, a neuromuscular training, has recently been used in many sports training fields.It is meaningful to present standards for effective VRT training. The expected outcome is that VRT training will be more effective than CRT training. The researcher is presenting the results based on 80% strength, and I want to add a part about the training period and gradual load increase. If it is difficult to provide evidence for this, it should be explained as a limitation of the study.

Respond: We appreciate Reviewer’s in-depth thinking and agree with the opinion.

With the available data from the included articles, we performed the meta-regression analyses for exercise characteristics to identify the effective exercise variable on improving maximum strength. We found that only load of VRT is correlated with the changes of maximum strength in untrained subjects.

 Meta-regression analysis.

Training experience

Exercise characteristics

Coefficient

Standard Error

T value

P value

Trained

Duration

.0823394

.1055939

0.78

0.452

Load

.7700836

.421567

1.83

0.095

Untrained

Duration

-1.214853

.5114492

-2.38

0.064

Load

-2.396898

.697879

-3.43

0.019*

* significant correlation between improved maximum strength and exercise characteristics.

Comment 2: In the discussion, it may be effective for muscle training that CRT use can give greater impetus to muscles. This should be explained.

Respond: Thank you for this comment. The discussion part was improved and irrelevant information was deleted. The revised Discussion part fully explained the role of VRT on improved maximum strength in subject with or without training experience. 

Comment3: The study was well written according to the procedure, but the conclusion was too simplified. It is necessary to present complex results based on valid reasons why VRT training is more effective

Respond: We sincerely thankful to the Reviewer for his/her appreciation of our writing. As suggested, we have completely revised our conclusions in the revised manuscript (Page 11).

Furthermore, the results were logically explained and possible reasons why VRT is effective than CRT in improving the maximum strength is now explained with suitable references. 

Reviewer 3 Report

Thanks for the opportunity to review this manuscript.

Recommendations:

The number of cited sources focused on the research topic needs to be expanded to correspond to a meta-analysis.

The impact of this research on previous research is not clear.

The conclusions should be extended to highlight the relevant aspects of the study.

Author Response

Reviewer 3

Dear Reviewer

All authors are thankful to you for your critical evaluation and meaningful comments on our manuscript. All comments helped us to improve the quality of our manuscript. We have taken special care to revise the entire manuscript. We have provided the additional details and data wherever necessary. All corrections in the revised manuscript were marked in red color. Now our detailed 'point-by-point' responses to each comment have been provided below. We believe the revised manuscript is more suitable for your valuable decision.

Comment 1: The number of cited sources focused on the research topic needs to be expanded to correspond to a meta-analysis.

Response: Thank you for your suggestion. We conducted the search process according to our search strategy. We believed that our search strategy had captured the available studies since we used broad search keywords.

Besides, we now emphasized the important studies and their findings in Introduction and Discussion sections.    

Comment 2: The impact of this research on previous research is not clear.

Response: The impact and importance of this study over previous studies is now explained in our revised manuscript. The main inputs can be found in the Introduction (Page 2 last paragraph) Discussion (Page 9 and 10) part of the revised manuscript.

Comment 3: The conclusions should be extended to highlight the relevant aspects of the study.

Response: Authors are thankful to the Reviewer for this useful comment. Now we have expanded our study conclusions and added research perspectives in the revised manuscript Page 11.   

Furthermore, the revised manuscript has been edited by MDPI English editing service.

Reviewer 4 Report

General comments: There are several awkward setences, which should be corrected: lines 69-70, 75-77, 81-82 (introduction) 87, 89-91, 95-96, 97, 124, 136-137, 144, 163-164 (discussion). Also, the terms "non-chain...., non-bungy were used" should be changed. 

The terms should be better defined (e.g. variable resistance training). Along the same lines, it is not clear what the authors analyzed, added chain and elastic resistance are different stimuli, however, the analysis combined these together. Maybe accomodating resistance would be a better term in case of this analysis.

The literature search process should be clear in the methods section (e. g. although it is indicated later, in the methods section, there is no mention about the date of the start of the search period). It is also not clear why the authors used the search term "power output", it seems to be irrelevant to the current work. The Flow chart is also confusing and should be clarified. What do the authors mean on "previous studies", "studies sought for retrieval" "studies not retrieved". New studies included and previous studies included are confusing, and it is not clear what they refer to. 

It would also be helpful to conduct a sub-analysis based on the length of intervention. It has been reported that the lenght of the training program is a major determinant of resistance training adaptations, so it may also contribute to the results in this analysis.

The discussion section should be rewritten. In its current form it is not specific enough and doesn't critically evaluate the existing data to compare and contrast with the current findings. On the other hand, it contains  discussion that is irrelevant to the present work. (e.g. speed of movement).

Author Response

Reviewer 4

Dear Reviewer

On behalf of our coauthors, I would like to appreciate the critical evaluation and meaningful comments on our manuscript. All comments helped us to improve the quality of our manuscript. With a special care we have revised our manuscript, and additional information or data were provided wherever necessary. All corrections in the revised manuscript were marked in red color. Now our detailed 'point-by-point' responses to each comment have been provided below. We believe the revised manuscript is more suitable for your valuable decision.

Comment 1: There are several awkward setences, which should be corrected: lines 69-70, 75-77, 81-82 (introduction) 87, 89-91, 95-96, 97, 124, 136-137, 144, 163-164 (discussion). Also, the terms "non-chain...., non-bungy were used" should be changed.

Response: We are apologizing for letting you read these sentences. As suggested, all listed sentences were carefully revised for better understanding. Then the revised manuscript was edited by MDPI English service.

Comment 2: The terms should be better defined (e.g. variable resistance training). Along the same lines, it is not clear what the authors analyzed, added chain and elastic resistance are different stimuli, however, the analysis combined these together. Maybe accomodating resistance would be a better term in case of this analysis.

Response: We agree with the reviewer's comment that accommodating resistance is a good term that was added in the introduction in Line48-49. Since variable-resistance training (VRT) is widely used and also previous meta-analyses described about VRT, we use the same term in our study.

                Besides, the unclear sentences and analyses in previous studies were clearly presented in the revised manuscript, Page 2. 

Comment 3: The literature search process should be clear in the methods section (e. g. although it is indicated later, in the methods section, there is no mention about the date of the start of the search period).

Response:  We are thankful to the reviewer for this comment and suggestions. As recommended, the literature search process has been revised in Methods section. The following subsections in the Methods further explained article identification, screening, selection, data extraction and quality assessment.

                We would like to bring to Reviewer’s kind notice that we have-not set the starting date for article search. We search all databases until 31 January 2022 without restricting the starting date.

Comment 4: It is also not clear why the authors used the search term "power output", it seems to be irrelevant to the current work.

Response: Authors are highly thankful to the reviewer for such an important suggestion. Now we have deleted the search term "power output" from our keywords.

Comment 5: The Flow chart is also confusing and should be clarified. What do the authors mean on "previous studies", "studies sought for retrieval" "studies not retrieved". New studies included and previous studies included are confusing, and it is not clear what they refer to.

Response: Authors are thankful to the Reviewer for this useful comment. We intended to say, previous meta-analysis that compared VRT and CRT effects on maximum strength included only 6 articles, which is relatively low. Here in our study, we included 8 additional studies which were not included in previous meta-analysis. Taken together, we included 14 articles and performed meta-analysis. However, we also feel that it is bit of confusing to understand the article numbers. Therefore, we have revised our PRISMA flow chart according to the latest PRISMA guidelines (2020) and final included articles remain same.

The terms "studies sought for retrieval" and "studies not retrieved" are adopted directly from the PRISMA 2020 template. In accordance with guidelines, usages of these terms are widely accepted for systematic reviews and meta-analysis.    

Comment 6: It would also be helpful to conduct a sub-analysis based on the length of intervention. It has been reported that the lenght of the training program is a major determinant of resistance training adaptations, so it may also contribute to the results in this analysis.

RespondWe appreciate Reviewer’s in-depth views about intervention length on maximum strength. We do agree with this point that length or duration of intervention is major determinant that could influence the VRT beneficial effects. To find out this phenomenon, we performed the meta-regression analysis for both duration and load of VRT, and found that duration was not correlated with changes of maximum strength. Interestingly, we found load of VRT is correlated with the improvement of maximum strength. Therefore, we subgrouped the studies based on training load and the results were clearly explained in the revised manuscript, Page 7 and 8.

                Meta-regression analysis to identify the effective exercise moderator on improved maximum strength. * Represents a significant correlation between maximum strength and exercise characteristics.

Training experience

Exercise characteristics

Coefficient

Standard Error

T value

P value

Trained

Duration

.0823394

.1055939

0.78

0.452

Load

.7700836

.421567

1.83

0.095

Untrained

Duration

-1.214853

.5114492

-2.38

0.064

Load

-2.396898

.697879

-3.43

0.019*

Comment 7: The discussion section should be rewritten. In its current form it is not specific enough and doesn't critically evaluate the existing data to compare and contrast with the current findings. On the other hand, it contains  discussion that is irrelevant to the present work. (e.g. speed of movement).

Response: Authors are thankful to the Reviewer for this vital comment. As mentioned, we have extensively revised our discussion part to emphasize the advantages of VRT over CRT based on available reports. The revised discussion and introduction parts specifically explained the influence of VRT on gaining of maximum strength. In addition, we further described the influence of VRT load on maximum strength in trained and untrained subjects. The irrelevant discussion part was removed and stick to the main scope of this study. 

Round 2

Reviewer 1 Report

Dear authors,

Thank you for considering my suggestions and incorporating them into the manuscript.

Below some more suggestions (minor details), with line indication.

The authors name abbreviations should be placed near the emails.

110 - Please review (publish ed”).

166 - Please review formula presentation quality, not clear for readers analysis.

167 - In the age column, some values with decimals and others not, please standardize. Also, (Squat+elastic & Squat +chain = diferent format, please standardize).

193 - Please improve the English. “were intercontinental, were published”.

203 - Italic, not according to other subtitles (for example line 191).

Although figures 2, 3 and 4 quality have improved, more quality is suggested aiming readers interpretation.

332 and 333 - Please consider elimination. It is suggested that the limitations paragraph is included in the discussion section, without subtopic and spaces.

345 - Please consider elimination.

378 - References format should be carefully reviewed and corrected according to the journal instructions for authors. Some examples: (378 – “øvretveit,”; “.J.” – in  many other cases the spaces should also be carefully reviewed). 398 “Strength & Conditioning Journal” and 399 “Strength Cond. J.”. These are only some examples, all references formats should be carefully reviewed, one other example is the article title in some refs in uppercase and in others in lowercase.

Congratulations for the research and keep up the good work.

Author Response

Review 1

Dear Reviewer, Thank you very much for your constructive comments and suggestions. Your suggestions further helped us to make minor changes to our manuscript. All corrections in the revised manuscript were marked in red color.

Our detailed 'point-by-point' responses to each comment have been provided below. We believe that the revised manuscript is more suitable for your valuable decision.

Comment 1: The authors name abbreviations should be placed near the emails.

Response: Thank you for your suggestion. Now we have added the abbreviated authors’ names next to their email ids respectively. These additions were done in the manuscript, Page 1, Lines 6-12.

Comment 2:110 - Please review (publish ed").

Response: Authors are thankful to the Reviewer for this comment. Now we have corrected this as ‘published’ in the revised manuscript, Page 3, Line 110.

Comment 3:166 - Please review formula presentation quality, not clear for readers analysis.

Response: We appreciate authors for this useful comment. Now we have revised the relevant sentences, improved the quality of equation presentation and explained the abbreviations in the equation. All corrections were done in the manuscript, Page 4, Lines 165-171.

Comment 4:167 - In the age column, some values with decimals and others not, please standardize. Also, (Squat+elastic & Squat +chain = diferent format, please standardize).

Response: Thanks to the Reviewer for this comment. As suggested, the decimal values in ‘age’ column have been amended and maintained the consistency. Moreover, the presentation of ‘squat+elastic’ and ‘squat+chain’ were also standardized in the Table 1. 

Comment 5:193 - Please improve the English. "were intercontinental, were published".

Response: We agree with the Reviewer’s opinion that presentation of this sentence was not smooth. Now we have revised this sentence for clarity in the revised manuscript, Page 7, Lines196 – 197.

Comment 6:203 - Italic, not according to other subtitles (for example line 191).

Response: Thanks for this correction. Now all subtitles in the Results section have been changed to italics for consistency.  

Comment 7:Although figures 2, 3 and 4 quality have improved, more quality is suggested aiming readers interpretation.

Response: Again, we are thankful to the Reviewer for this suggestion. As recommended, we have now further improved the quality of Figure 2, 3 and 4. In fact during file conversion (Word to PDF), the quality of Figures is getting low. The Word file remain shows the high quality Figures. However, we can provide separate Figures during proof correction, if necessary.

Comment 8:332 and 333 - Please consider elimination. It is suggested that the limitations paragraph is included in the discussion section, without subtopic and spaces.

Response: We agree with your suggestion. Now we have deleted the ‘Limitations’ subtitle, which resulted ‘limitations’ are part of discussion at the end. 

Comment 9:345 - Please consider elimination.

Response: According to the suggestion, we no deleted the sentence in the Conclusion part.

Comment 10:378 - References format should be carefully reviewed and corrected according to the journal instructions for authors. Some examples: (378 – "øvretveit,"; ".J." – in  many other cases the spaces should also be carefully reviewed). 398 "Strength & Conditioning Journal" and 399 "Strength Cond. J.". These are only some examples, all references formats should be carefully reviewed, one other example is the article title in some refs in uppercase and in others in lowercase.

Response: We are apologizing for the poor formatting of the reference list. This poor format is due to the old version of our ‘reference manager’. We now used the latest version of EndNote to format the reference style. In addition, each reference was manually checked and corrected according to the journal guidelines. The revised format is much more suitable for the journal style.

Reviewer 3 Report

The authors improved the article according with the recommendations. 

Author Response

Dear Editor, Thanks a lot for your approval.

We appreciate your time.

Best regards

Reviewer 4 Report

Corrections needed:

Line 78-population

Line 199-...included chain and elastic.... (?) Perhaps, chain and elastic resistance

Line 251 ...overall VRT was better...   This statement is not supported by the result, because the effectiveness of the intervention depends on the load and the training state of the subjects, so the reviewer suggests to exclude or replace the word "overall" in this sentence.

Line 260 ...lift weights in the centripetal phase... It is not clear what this statement refers to in this context.

Line 269 ...activation of the active muscle... This statement is awkward and should be corrected.

Lines 271-272 ...when the lines of squat and bench press rise... This statement does not make any sense and should be corrected.

Line 281 perhaps less than 80% 1RM loads

Lines 297-298 Muscle adaptation.... This sentence is very general and not very scientific, so it should be corrected.

Lines 304-305 ..it becomes complicated to increase other muscle sizes and strength. This statement does not make any sense and should be corrected.

Line 306 ...adapt the muscles to the neuromuscular system... This statement does not make any sense and should be corrected.

Line 327 ...output of the force... simply "force output"

Author Response

Review 4

Dear Reviewer,

Thank you again for your time and valuable comments on our manuscript. According to your suggestions, now we carefully revised our manuscript for further improvement of quality presentation. All the corrections in the revised manuscript were marked in red color. Our detailed 'point-by-point' responses to each of your comment have been provided below. We believe that the revised manuscript is more suitable for your valuable decision.

Comment 1: Line 78-population

Response: Thanks for the suggestion. We have now corrected the spelling of ‘population’ in the revised manuscript, Line 78. 

Comment 2: Line 199-...included chain and elastic.... (?) Perhaps, chain and elastic resistance

Response: We are thankful to you for this good suggestion. As recommended, we have now included the ‘resistance’ after ‘chain and elastic’ in Line 203.

Comment 3: Line 251 ...overall VRT was better...   This statement is not supported by the result, because the effectiveness of the intervention depends on the load and the training state of the subjects, so the Reviewer suggests to exclude or replace the word "overall" in this sentence.

Response: We do agree with the Reviewer’s opinion that ‘overall’ may not be suitable to emphasize the beneficial effects of VRT. Therefore, we have deleted the ‘overall’ from our statement in the Discussion section first paragraph, which gives clear meaning.

Comment 4: Line 260 ...lift weights in the centripetal phase... It is not clear what this statement refers to in this context.

Response: Thanks for this critical comment. The word ‘centripetal’ has been modified to ‘concentric’, which gives direct and clear meaning of this statment. Please see the change in the manuscript, Line 259.

Comment 5: Line 269 ...activation of the active muscle... This statement is awkward and should be corrected.

Response: Now, we have changed the ‘active muscle’ to ‘vastus lateralis’ as mentioned in the original article. This correction in the revised manuscript is visible in Page 10, Line 268.

Comment 6: Lines 271-272 ...when the lines of squat and bench press rise... This statement does not make any sense and should be corrected.

Response: Apologizing for the unclear sentence in the previous version of the manuscript. Now we have revised the statement for clarity (Page 10, Line 270).

 Comment 7: Line 281 perhaps less than 80% 1RM loads

Response: Yes, this is less than 80% 1RM loads. Thank you for mentioning this point. Now we have amended this in our revised manuscript, Line 279.

Comment 8: Lines 297-298 Muscle adaptation.... This sentence is very general and not very scientific, so it should be corrected.

Response: Thanks for the comment. Now this sentence has been revised and presented scientifically in the manuscript, Lines 295-296.

Comment 9: Lines 304-305 ..it becomes complicated to increase other muscle sizes and strength. This statement does not make any sense and should be corrected.

Response: We are thankful to the Reviewer for printout this statement. We have deleted this unclear sentence without changing the original meaning.

Comment 10:Line 306 ...adapt the muscles to the neuromuscular system... This statement does not make any sense and should be corrected.

Response: We appreciate Reviewer for the careful revision of the revised manuscript. Now we have deleted this unclear sentence in the revised manuscript.

Comment 11:Line 327 ...output of the force... simply "force output"

Response: Authors are thankful to the Reviewer for this suggestion. As suggested, now we have changed ‘output of the force’ to ‘force output’ in the revised manuscript, Page 11, Line 323.
